# Ultrasound-Guided Anterior Mediastinotomy: A Feasible Tool for Critical Lymphoma Patients

**DOI:** 10.3390/healthcare9060770

**Published:** 2021-06-21

**Authors:** Roberto Cascone, Annalisa Carlucci, Gaetana Messina, Antonio Noro, Mary Bove, Giovanni Natale, Caterina Sagnelli, Giorgia Opromolla, Mario Martone, Carlo Santoriello, Elena Santoriello, Pasquale Verolino, Massimiliano Creta, Giovanni Vicidomini, Alfonso Fiorelli, Mario Santini, Antonello Sica

**Affiliations:** 1Thoracic Surgery Unit, University of Campania Luigi Vanvitelli, 80131 Naples, Italy; rob.cascone@libero.it (R.C.); annalisacarlucci88@gmail.com (A.C.); adamessina@virgilio.it (G.M.); ant.noro@gmail.com (A.N.); bovemary10@gmail.com (M.B.); dott.natale.giovanni@gmail.com (G.N.); giorgia.opromolla@studenti.unicampania.it (G.O.); mario.martone3@gmail.com (M.M.); santoriellocarlo@gmail.com (C.S.); santorielloelena@gmail.com (E.S.); giovanni.vicidomini@unicampania.it (G.V.); alfonso.fiorelli@unicampnaia.it (A.F.); mario.santini@unicampania.it (M.S.); 2Department of Mental Health and Public Medicine, University of Campania Luigi Vanvitelli, 80131 Naples, Italy; 3Plastic Surgery Unit, Multidisciplinary Department of Medical-Surgical and Dental Specialties, Luigi Vanvitelli University of Campania, 80131 Naples, Italy; pasquale.verolino@unicampania.it; 4Department of Neurosciences, Reproductive Sciences and Odontostomatology, University of Naples “Federico II”, 80131 Naples, Italy; max.creta@gmail.com; 5Department of Precision Medicine, University of Campania Luigi Vanvitelli, 80131 Naples, Italy; antonello.sica@fastwebnet.it

**Keywords:** mediastinal lymphoma, mediastinotomy, Chamberlain procedure, ultrasonography

## Abstract

A significant part of all neoplasms growing in anterior mediastinum are lymphomas (25%). Achieving a correct diagnosis and a clear definition of a lymphoma’s subtype is crucial for beginning chemotherapy as soon as possible. However, most patients present a large mediastinal mass that compresses vessels and airway, with serious cardiorespiratory repercussions. Therefore, having multiple tools available to biopsy the lesion without worsening morbidity becomes fundamental. Patients enrolled in this study were unfit for a surgical biopsy in general anesthesia and the need to begin chemotherapy as fast as possible prompted us to avoid percutaneous fine needle aspiration to prevent diagnostic failures. Our observational study included 13 consecutive patients with radiological findings of anterior mediastinal mass. Ultrasonography was performed directly in the theatre to mark the lesion and to localize vessels and vascularized neoplastic tissue. Open biopsy was carried out in spontaneous breathing with a laryngeal mask and with short-acting medications for a rapid anesthesia, performing an anterior mediastinotomy. The mean operative time was 33.4 ± 6.2 min and spontaneous respiration was maintained throughout the procedure. No complications were reported. All patients were discharged in the first or second postoperative day after a chest X-ray (1.38 ± 0.5 days). The diagnostic yield of this approach was 100%. With the addition of ultrasonography right before the procedure and with spontaneous breathing, anterior mediastinotomy still represents a useful tool in critical patients that could hardly tolerate a general anesthesia. The diagnostic yield is high, and the low postoperative morbidity allows a rapid onset of chemotherapy.

## 1. Introduction

Mediastinum can be the site of various lesions, either malignant or benign, that require a proper histological or cytological diagnosis. The classical surgical procedure performed to obtain a sample of the mediastinal mass is the anterior mediastinotomy (AM), otherwise known as the Chamberlain procedure [1]. It is a minimally invasive operation that enables exploration of anterior mediastinum and biopsy of neoplasm growing in this compartment.

In the last decades, further biopsy techniques arose to achieve both an adequate sampling of the lesion and a lower incidence of complications, such as fine needle aspiration cytology (FNAC) or video-assisted thoracoscopic surgery (VATS) [2].

More than two-third of all mediastinal masses are benign but those growing in the anterior mediastinum are often neoplasms. [3] Anterior mediastinal masses are a wide spectrum of lesions with different etiology, pathogenesis, and clinical behavior, so their management by the clinician could be difficult [4]. One of the most frequent etiologies discovered in these masses is lymphoma, with an average incidence of 25%. Although FNAC is a valuable tool for the diagnosis of lymphoma, thanks to the minimally invasive approach, it is not considered as first choice by the recent guidelines; the pathologic diagnosis of lymphoma can be difficult; therefore, a surgical biopsy is often preferred to obtain an adequate amount of tissue for diagnosis and definition of lymphoma subtypes [5,6,7,8,9,10,11,12,13,14].

The role of VATS is crucial because it combines the low invasiveness with the chance of getting a histological sample; however, VATS biopsy of mediastinal mass requires general anesthesia and intubation with correct lung exclusion, so not many patient results fit for this type of procedure due to the large neoplasm compressing mediastinal.

We explored the feasibility of biopsies performed in an old fashioned anterior mediastinotomy but carried out in sedation with spontaneous respiration; to minimize post-operatory complications, we aimed to speed up the surgical procedure without increasing surgical complications, so an accurate localization of mass with ultrasonography was done right before the procedure.

We also applied the most recent techniques of loco-regional analgesia to achieve an optimal pain management and shorten the length of stay.

The aim of our study was to evaluate if the combination of all these approaches was both effective in reaching a correct diagnosis and safer for a critical patient with a malignant mediastinal mass.

## 2. Materials and Methods

Our study population included 13 consecutive patients with radiological findings of large mediastinal mass in chest X-ray.

The characteristics of the study population are summarized in Table 1.

A high-resolution CT-angiography with narrow slice width was performed in all patients to assess both anatomical and vascular condition. Pre-operatory evaluation was carried out with cardiological examination, including electrocardiogram (ECG) and echocardiography. Functional respiratory tests, including standard spirometry, 6-min walking test, and arterial blood gas test, were also executed. Eventually, an anesthesiologist assessed the overall surgical risk addressing patients in the correct ASA physical status.

### Ultrasonography and Surgical Procedure

Ultrasonography (US) evaluation was carried out directly in the operatory theatre, a convex transducer to properly localize the lesion. When an adequate visualization of the mass was achieved, the site of incision was marked with a dermographic pen. To obtain an optimal pain control in the postoperative period, a US-guided Erector Spinae Plane (ESP) block was performed with a bolus of ropivacaine 1% and dexamethasone [15]. Patients were then positioned supine, and a pillow was placed under the thorax to facilitate the exposure of the mass. Short-acting medications were used to achieve a rapid anesthesia and a spontaneous breathing was maintained during the entire procedure; to guarantee a proper ventilation, a laryngeal mask was placed by the anesthesiologist. A single 2 cm transverse incision was performed in the intercostal space, following the cutaneous landmark obtained with the US. The chest layer was then opened sequentially until the appearance of the mass, paying attention to not incise the mediastinal pleura; unlike the classical Chamberlain procedure, there was no need for resection of the costal cartilage because the US allowed us to reach with pinpoint accuracy the lesion (Figure 1 and Figure 2).

The neoplasm was then sampled with biopsy forceps and sent to the pathologist, ensuring an adequate amount of tissue for the immunohistochemistry. Accurate hemostasis was then carried out before closure of the chest. The patient was brought back to the ward after a short observation period in the theatre and a chest X-ray was performed to rule out a pneumothorax. Post-operative pain was also quantified using a visual analogue scale (VAS) with 11-point levels, ranging from 0 (no pain) to 10, right after the surgical procedure, after 12 h and after 24 h.

## 3. Results

The mean operative time was 33.4 ± 6.2 min and spontaneous respiration was maintained throughout the procedure. Arterial blood gas tests were done throughout the procedure as well as continuous oxygen saturation monitoring being demonstrated. No complications were observed, and all patients were discharged in the first or second postoperative day after a chest X-ray (1.38 ± 0.5 days). Postoperative pain detected was 2.92 ± 1.11 at the end of procedure, 3.15 ± 0.55 12 h after surgery, and 2.69 ± 0.48 24 h after surgery. 

Thanks to ESP block, pain management during recovery time was carried out with only intravenous Paracetamol; avoiding opioid drugs allowed an early mobilization and feeding of patients within 12 h after procedure as well as a fast discharge (mean 1.38 ± 0.5 days) (Table 2).

All samples sent to pathologist were sufficient to reach diagnosis of lymphoma with accurate definition of subtypes. We observed Primary mediastinal/thymic large B-cell lymphoma in six patients, Diffuse large B-cell lymphoma (DLBCL), NOS (Germinal centre B-cell type) in 2 cases, Diffuse large B-cell lymphoma (DLBCL), NOS (Activated B-cell type) in one case, and Nodular Sclerosis Classical Hodgkin lymphoma (NSCHL) in four cases.

Then, the patients were addressed to proper chemotherapy and/or radiotherapy within 30 days after the biopsy. Two weeks after surgery, patients were examined in the outpatient department for stitches removal and radiological control; chest X-ray excluded pneumothorax in all cases. 

## 4. Discussion

Histological diagnosis of a mediastinal mass could often be challenging for a pathologist, especially when it comes to lymphoproliferative lesions. From the clinical point of view, we discriminate between two large families of lymphoma: the classical Hodgkin lymphoma (CHL) and the non-Hodgkin lymphoma (NHL). Therapeutic approach and prognosis are very different between these two types of neoplasm and even between their various histological subtypes [15,16,17,18,19,20,21,22,23,24,25,26,27,28,29,30,31]. The need of a correct and rapid diagnosis is therefore crucial for channeling patients to a proper chemotherapy, improving survival rates. Typical patients with primary mediastinal lymphoma refer to a physician with symptoms of superior vena cava syndrome, due to tumor compression of mediastinal structures. Other clinical manifestations are compression of recurrent laryngeal nerve with dysphonia, dyspnea, and retrosternal chest pain. These symptoms are caused by a large bulky disease that develops in the mediastinum, compressing or invading adjacent structures. A chest X-ray can easily show the large mediastinal radiopacity but a CT scan or a magnetic resonance of the thorax is often required to highlight the involvement of vessels [32,33,34,35]. Cardiorespiratory conditions of patients with large mediastinal lymphoma are, therefore, critical and their management could be hard in an operative setting. Various biopsy techniques are commonly used to achieve diagnosis, ranging from minimally invasive ones to surgical procedures. The pros and cons of each procedure are summarized in Table 3.

The CT-guided FNAB is surely the most used technique in mediastinal masses thanks to low rates of complications and fast execution, but it provides only a cytological sample, making diagnosis uncertain.

Several studies investigated the feasibility of CT-guided core-needle biopsy in mediastinal masses as a good alternative to open biopsy, allowing collection of histological samples compared to FNAC. Petranovic et al. [36] based on core-needle biopsy reported acceptable diagnostic yield (77%) for anterior mediastinal masses but it does not differentiate between thymic or lymphoproliferative neoplasms. Iguchi et al. [37] demonstrated that a real time CT fluoroscopy-guided core needle biopsy can be more efficient in terms of diagnostic yield; however, this comes at the expense of a higher radiation dose to the patient. 

The most common adverse event reported during a percutaneous procedure on mediastinum are pneumothorax and bleeding that could spontaneously resolve [38].

In our opinion, although these complications can be easily resolved, they should still be avoided because they could slow down the therapeutic path or worsen the status of a critical patient with mediastinal lymphoma. Tacconi et al. [39] described a useful approach to investigate lesions in the anterior mediastinum, using a minimally uniportal VATS in an awake setting. The diagnostic yield reported in their work was 100% with a low morbidity rate. However, this approach required the placement of a drainage tube, a factor that could affect the post-operative pain due to intercostal nerve compression. 

We wanted to step up this procedure cutting out the opening of the pleural cavity, thus avoiding the chest drain and improving the patients’ tolerance to surgical biopsy. When carried out with a rigorous technique, the anterior mediastinotomy is a rapid procedure that ensures an adequate amount of tissue with few or no complications. According to our experience, the main factor that sped up the operative time was the use of US to localize the lesion right before the procedure.

Ultrasonography is a well described tool when talking about image-guided percutaneous needle-biopsy. Its use in the diagnostic process of mediastinal lesions is widespread, thanks to the possibility to perform it in various settings, from radiology department to the bed of patient. Compared to the CT scan, it does not use ionizing radiation for the benefit of the patient. Even if is operator dependent methodology, US guaranteed a rapid localization of the mass in our operatory theatre and color Doppler allowed us to correctly identify vessels and vascularized tumoral tissue to avoid intraoperative complications.

Anesthesiologic management of mediastinal surgery is complex, due to neoplastic compression of vessels or airway [40,41,42]. Avoiding general anesthesia and orotracheal intubation, the morbidity rate was low in our study population, with only a few cases of adverse events that were easily managed with a conservative approach or minimally invasive drainage. Moreover, the use of ESP block demonstrated that postoperative pain was noticeably low, with positive effects on patients’ compliance to early mobilization and feeding after surgery [43]. This fast-track surgery approach appeared to be effective in reducing the postoperative length of stay, enabling the starting of chemotherapy or radiotherapy as fast as possible.

## 5. Conclusions

In the era of minimally invasive approaches, anterior mediastinotomy still represents a useful tool when dealing with critical patients that could hardly tolerate a general anesthesia. With the addition of ultrasonography right before procedure to proper identify mass and vessels, this methodology can be carried out rapidly and with few or no perioperative complications, with a diagnostic yield of 100% thanks to the large amount of tissue obtained.

However, this study has several limitations, being an observational monocentric study without randomization; so, more studies are required to strengthen the results we obtained. In addition, in the case of median lymphomas, very often we are dealing with Hodgkin’s lymphoma. In the case of node biopsy with HL, sometimes false negative results are found, and this represents a possible limitation of the method with respect to surgical biopsy.

## Figures and Tables

**Figure 1 healthcare-09-00770-f001:**
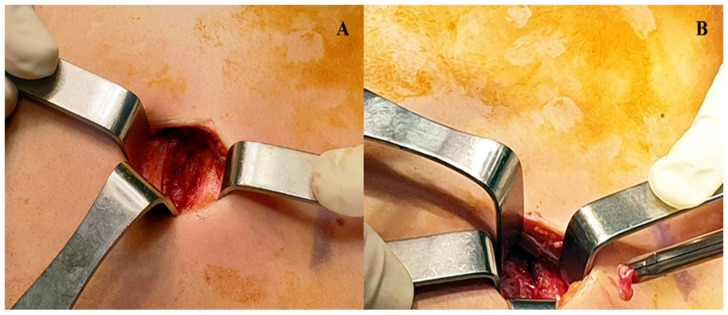
Anterior mediastinotomy (**A**) and sampling of mediastinal mass with biopsy forceps (**B**).

**Figure 2 healthcare-09-00770-f002:**
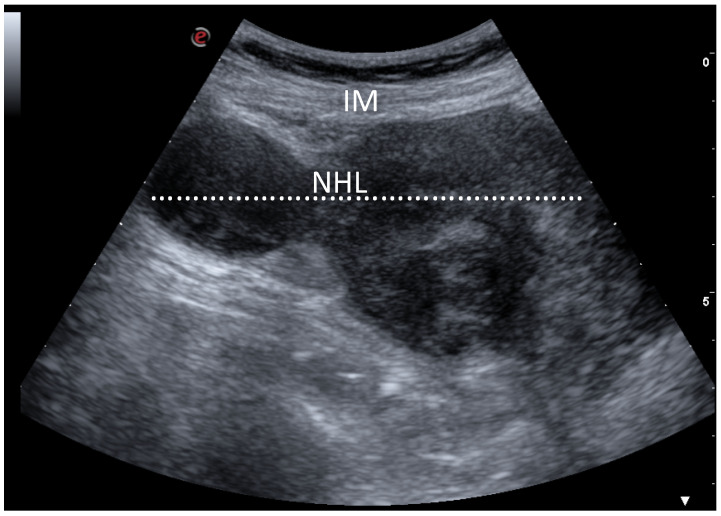
Detection of anterior mass by ultrasound. Ultrasound with a convex transducer placed at the level of the second intercostal space showed the presence of a large mass (70 mm size) with heterogeneous echogenicity (dotted line). The diagnosis was non-Hodgkin’s lymphoma. IM: Intercostal muscle, NHL: Non-Hodgkin’s lymphoma.

**Table 1 healthcare-09-00770-t001:** Characteristics of the study population.

Variables	
Number of patients	13
Age, years (M ± SD)	55.69 ± 19.11
Male sex, N° (%)	8 (61.5)
Comorbidity, N° (%):	
COPD	6 (46.2)
Cardiac	6 (46.2)
Diabetes mellitus	5 (38.5)
Functional tests (M ± SD)	
FEV1	81.07 ± 11.25
6 mwt (m)	370.76 ± 72.27
EF	48.84 ± 9.38

COPD: Chronic Obstructive Pulmonary Disease; FEV1: Forced Expiratory. Volume in the 1st second; 6 mwt: six minutes walking test; EF: Ejection Fraction.

**Table 2 healthcare-09-00770-t002:** Perioperative results.

Variables		*p* Value
Operative time	33.46 ± 6.21	<0.0001
(M ± SD)		
VAS (M ± SD)		
After surgery	2.92 ± 1.11	<0.0001
12 h	2.69 ± 0.48	<0.0001
24 h	3.15 ± 0.30	<0.0001
Length of stay	1.38 ± 0.50	<0.0001

VAS: Visual Analogue Scale.

**Table 3 healthcare-09-00770-t003:** Different techniques to sample mediastinal mass.

Technique	Pros	Cons
CT guided or US—FNAB	- No need for operating room - No need for anesthetist - Spontaneous breathing without any sedation - No need for recovery - Very low risk of complication	Sample insufficient for a definitive diagnosis
CT or US guided Core-needle biopsy	- No need for operating room - No need for anesthetist - Spontaneous breathing without any sedation - No need for recovery More materials for diagnosis than FNAB	- Higher risk of complication compared to FNAB CT guided - Sample insufficient for a definitive diagnosis
VATS biopsy	- Adequate sample for a diagnosis	- Needing recovery - Needing operating room - Needing general anesthesia with one-lung ventilation - Chest drainage after the procedure
Standard mediastinotomy (Chamberlain technique)	- Adequate sample for diagnosis - No need for one-lung ventilation - No drainage after the procedure	- Needing recovery - Needing operating room - Needing general anesthesia - Resection of cartilage rib
Present technique (US associated with anterior mediastinotomy)	- Adequate sample for diagnosis - No need for general anesthesia - No drainage - No drainage after procedure - No resection of cartilage rib	- Needing recovery - Needing operating room - Needing anesthetist

CT: Computed-tomography, US: ultrasound, FNAB: guided Fine-needle aspiration biopsy, VATS: Video-assisted thoracoscopic surgery.

## Data Availability

Data sharing not applicable.

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
