# Peer review of "Ultrasound-Guided Anterior Mediastinotomy: A Feasible Tool for Critical Lymphoma Patients"

_healthcare, 2021, doi:10.3390/healthcare9060770_

Round 1

Reviewer 1 Report

Article entitled ” Ultrasound-guided anterior mediastinotomy: a feasible tool for critical lymphoma patients” presents very interesting possibilities of a quick diagnosis of lymphomas. The authors used a valid methodology and presented the correct conclusions from the analysis.

I have a minor comments:

  • What subtypes of lymphoma did you find? Including this information would enrich the article.
  • In the case of mediastinal lymphomas very often we have to deal with Hodgkin lymphoma. In the case of biopsy of nodes with HL, sometimes false-negative results are found.I think that this should be taken into account in the possible limitations of the method compared to the surgical biopsy.

I recommend an article for publication in “Healthcare”.

Author Response

To the Editor in Chief of HEALTHCARE

Dear Editor,

We re-submit our paper entitled: “Ultrasound-guided anterior mediastinotomy: a feasible tool for critical lymphoma patients”, healthcare-1233152.

The manuscript has been extensively revised and edited to accommodate all the criticisms and suggestions of the reviewers, whom we thank for their help to improve this narrative review.

According to the suggestion of the reviewer the following changes have been made.

Reviewer 1

Article entitled ” Ultrasound-guided anterior mediastinotomy: a feasible tool for critical lymphoma patients” presents very interesting possibilities of a quick diagnosis of lymphomas. The authors used a valid methodology and presented the correct conclusions from the analysis.

I have a minor comments:

Point 1 What subtypes of lymphoma did you find? Including this information would enrich the article.

Response 1: We accepted the suggest of the reviewer and modified the new manuscript accordingly.

Point 2: In the case of mediastinal lymphomas very often we have to deal with Hodgkin lymphoma. In the case of biopsy of nodes with HL, sometimes false-negative results are found.I think that this should be taken into account in the possible limitations of the method compared to the surgical biopsy.

Response 2: We accepted the suggest of the reviewer and modified the new manuscript accordingly.

Reviewer 2

In an observational study of 13 patients the authors present a minimally invasive approach (anterior mediastinotomy assisted by immediate ultrasonography) as a useful tool in the diagnostic evaluation of mediastinal masses in certain scenarios.

Specific Points of Criticism and Suggestions for Alterations:

Point 1 While the English is generally good, here and there are some minor mistakes (e.g. „desamethasone“, „maintened“, „patients … refer to clinican“ and others elsewhere) which should be corrected by thorough editing of the text.

Response 1: We accepted the suggest of the reviewer and modified the new manuscript accordingly.

Point 21 Figure 2:  It might be useful to employ some arrows to indicate what the reader is supposed to appreciate in this figure (and should be explained accordingly in the legend).

Response 2: We accepted the suggest of the reviewer and modified the new manuscript accordingly.

Point 3 Line 175 (Discussion):  Instead of citing 32 references (many of which are definitely outdated and simply too „old“) it might be more useful to cite a few but more recent references, like for example relevant overview articles.

Response 3: We accepted the suggest of the reviewer and modified the new manuscript accordingly.

Point 4 Summary Table:  A table summarizing the main aspects of the present technique would be useful and a good take-home message – possibly as a comparison table listing various different techniques suitable for different scenarios with their advantages and disadvantages, including possible adverse events.

Response 4: We accepted the suggest of the reviewer and modified the new manuscript accordingly.

Point 5 List of references:  There are quite a lot of „Sica-references“ (n = 15) which seems excessive and the authors should avoid too many self-citations. Furthermore, 64 references for such a short paper seems too many. As mentioned above, many „old“ references going far back could be deleted (as this is also not a historical review article).

Response 5: We accepted the suggest of the reviewer and modified the new manuscript accordingly.

We thank again the Editor and the Reviewers for the suggestions given that have allowed us to improve our paper.

The manuscript has been read and approved by all the authors and has not been submitted for publication to other journals. We also declare that no author has conflict of interest in connection with this paper.

Neither the manuscript nor part of it has been published or is under consideration for publication elsewhere.

We sincerely hope that the enclosed manuscript can be accepted for publication in the " HEALTHCARE”.

Sincerely yours,

 Prof.ssa Caterina Sagnelli

Reviewer 2 Report

In an observational study of 13 patients the authors present a minimally invasive approach (anterior mediastinotomy assisted by immediate ultrasonography) as a useful tool in the diagnostic evaluation of mediastinal masses in certain scenarios.

Specific Points of Criticism and Suggestions for Alterations:

(1) While the English is generally good, here and there are some minor mistakes (e.g. „desamethasone“, „maintened“, „patients … refer to clinican“ and others elsewhere) which should be corrected by thorough editing of the text.

(2) Figure 2:  It might be useful to employ some arrows to indicate what the reader is supposed to appreciate in this figure (and should be explained accordingly in the legend).

(3) Line 175 (Discussion):  Instead of citing 32 references (many of which are definitely outdated and simply too „old“) it might be more useful to cite a few but more recent references, like for example relevant overview articles.

(4) Summary Table:  A table summarizing the main aspects of the present technique would be useful and a good take-home message – possibly as a comparison table listing various different techniques suitable for different scenarios with their advantages and disadvantages, including possible adverse events.

(5) List of references:  There are quite a lot of „Sica-references“ (n = 15) which seems excessive and the authors should avoid too many self-citations. Furthermore, 64 references for such a short paper seems too many. As mentioned above, many „old“ references going far back could be deleted (as this is also not a historical review article).

Author Response

(The authors gave the same response as above.)
